# Robust and Scalable Angiogenesis Assay of Perfused 3D Human iPSC-Derived Endothelium for Anti-Angiogenic Drug Screening

**DOI:** 10.3390/ijms21134804

**Published:** 2020-07-07

**Authors:** Vincent van Duinen, Wendy Stam, Eva Mulder, Farbod Famili, Arie Reijerkerk, Paul Vulto, Thomas Hankemeier, Anton Jan van Zonneveld

**Affiliations:** 1Einthoven Laboratory for Vascular and Regenerative Medicine, Department of Internal Medicine (Nephrology), Leiden University Medical Center, 2333ZA Leiden, The Netherlands; w.stam@lumc.nl (W.S.); e.mulder01@gmail.com (E.M.); 2Department of Analytical BioSciences and Metabolomics, Division of Systems Biomedicine and Pharmacology, Leiden University, 2333CC Leiden, The Netherlands; hankemeier@lacdr.leidenuniv.nl; 3Ncardia, 2333 BD Leiden, The Netherlands; f.famili@ncardia.com (F.F.); a.reijerkerk@ncardia.com (A.R.); 4Mimetas, 2333 CA Leiden, The Netherlands; p.vulto@mimetas.com

**Keywords:** angiogenesis, drug screening, microfluidics, iPSC, endothelial cells

## Abstract

To advance pre-clinical vascular drug research, *in vitro* assays are needed that closely mimic the process of angiogenesis *in vivo*. Such assays should combine physiological relevant culture conditions with robustness and scalability to enable drug screening. We developed a perfused 3D angiogenesis assay that includes endothelial cells (ECs) from induced pluripotent stem cells (iPSC) and assessed its performance and suitability for anti-angiogenic drug screening. Angiogenic sprouting was compared with primary ECs and showed that the microvessels from iPSC-EC exhibit similar sprouting behavior, including tip cell formation, directional sprouting and lumen formation. Inhibition with sunitinib, a clinically used vascular endothelial growth factor (VEGF) receptor type 2 inhibitor, and 3-(3-pyridinyl)-1-(4-pyridinyl)-2-propen-1-one (3PO), a transient glycolysis inhibitor, both significantly reduced the sprouting of both iPSC-ECs and primary ECs, supporting that both cell types show VEGF gradient-driven angiogenic sprouting. The assay performance was quantified for sunitinib, yielding a minimal signal window of 11 and Z-factor of at least 0.75, both meeting the criteria to be used as screening assay. In conclusion, we have developed a robust and scalable assay that includes physiological relevant culture conditions and is amenable to screening of anti-angiogenic compounds.

## 1. Introduction

Angiogenesis, the growth of new blood vessels from pre-existing vasculature, plays a fundamental role in both health and disease [1]. For the discovery of new drug targets that target the angiogenesis process, drug research heavily relies on *in vitro* models. However, currently used *in vitro* models have limited translatability to the *in vivo* situation [2,3]. To meet the demands of pre-clinal vascular drug research, improved *in vitro* models of angiogenesis are required: assays that are amenable to high-throughput screening, with a scalable and robust endothelial cell (EC) source in a more physiologically relevant cellular micro-environment [4,5].

Within the last decade, significant progress has been made to increase the translational value of *in vitro* models of angiogenesis. For example, ECs embedded in three-dimensional scaffolds such as fibrin and collagen gels show an increased physiologically relevant phenotype, including the presence of tip and stalk cells. The tip cells are able to degrade the extracellular matrix, while the stalk cells form lumen [6,7]. However, important cues from the cellular microenvironment, such as biomolecular gradients and flow, are still lacking.

The use of microfluidic cell culture platforms can further increase the physiological relevancy of *in vitro* models, as perfusion of culture media induces shear stress in the ECs and allows spatial- temporal control over biomolecular gradients [8]. This has resulted in angiogenic sprouting models with increased physiological relevancy over traditional 2D and 3D cell culture methods. For example, it enables the formation of gradients in combination with a 3D scaffold, the stratification of cells to force polarization and the possibility to apply luminal perfusion [9,10,11,12,13,14,15]. However, most platforms lack the required throughput and scalability in order to be amenable to drug research in general, and drug screening in particular [16]. Furthermore, many of these platforms require the end-user to microfabricate their devices prior to use [17]. This not only requires manufacturing equipment and technical knowledge; it also limits the level of quality control and negatively affects the reproducibility and standardization [18].

To date, primary human ECs such as human umbilical vein endothelial cells (HUVECs) remain the most widely used cell source to model angiogenesis *in vitro* [4]. However, while HUVECs have the advantage of being widely available, being robust in their performance and being expandable up to a certain level, the performance of these primary endothelial cells can widely vary from donor to donor, limiting their use for high throughput assays over longer times. ECs derived from induced pluripotent stem cells (iPSC) are a promising alternative. Since iPSCs are self-renewing, they can be expanded in nearly unlimited quantities, and yield endothelial cells with phenotypic properties highly like primary endothelial cells. In addition, iPSCs are amenable to precise gene editing, allowing the introduction of a fluorescent phenotypic reporter that facilitates high throughput imaging. Together, these properties make iPSC-EC a highly favorable source for *in vitro* screening models assays of endothelial function [19].

Recently, we introduced a platform technology that comprises 40 microfluidic chips patterned underneath a microtiter plate [13,20]. We showed formation of a small EC vessel structures with perfused lumen, grown against an extracellular matrix. In this model, like primary ECs, iPSC-ECs, showed important aspects of angiogenic sprouting, including the differentiation into tip cells that display their characteristic filopodia and the trailing stalk cells forming lumen [13]. Both directional and repetitive sprouting was observed, which was an indication that iPSC-ECs can sense the imposed vascular endothelial growth factor (VEGF) gradient and suggested that this could be used to screen for anti-angiogenic properties.

Here, we investigate whether this platform is suitable for anti-angiogenic screening. We compared the angiogenic response of iPSC-ECs with HUVEC and study the effect of sunitinib and 3-(3-pyridinyl)-1-(4-pyridinyl)-2-propen-1-one (3PO), two anti-angiogenic compounds that have been shown to reduce angiogenesis *in vitro* as well as *in vivo* [21]. Sunitinib is a clinically available tyrosine kinase inhibitor that targets the vascular endothelial growth factor receptor 2 (VEGFR2) and platelet-derived growth factor receptor beta (PDGFRβ). As a glycolysis inhibitor, 3PO targets 6-phosphofructo-2-kinase/fructose-2,6-bisphsphatase isozyme 3 (PFKFB3). We calculated the signal window, Z’ and assay variability window for several metrics to quantify the assay performance and assess its suitability for drug screening.

## 2. Results

The platform consists of 40 individually addressable microfluidic units (Figure 1a,b), in which 40 perfused microvessels are cultured against a patterned collagen-I gel. First, we validated the usage of iPSC-ECs in our angiogenesis model [20]. Briefly, collagen-1 gel precursor is patterned inside the chips by a surface tension technique named phaseguiding [22]. After polymerization, fibronectin coating was added in the adjacent channel. Cells were seeded in this coated channel and after they were adhered, additional culture media was added to the wells addressing the lumen. The device was placed on an interval rocker platform set at a 7 degrees inclination angle and an 8 min interval to induce flow through passive levelling between the reservoirs, and thus sustain gradients of angiogenic factors (Figure 1c). Formation of confluent iPSC-EC microvessels was realized after 2 days (Figure 1d), after which angiogenesis was triggered by a gradient of angiogenic growth factors (50 ng/mL VEGF + 500 mM, Sphingosine-1-Phosphate (S1P) + 2 µg/mL phorbol 12-myristate 13-acetate (PMA) for another two days. This shows that iPSC-ECs form angiogenic sprouts including tip cells as well as their characteristic filopodia and stalk cells that form perfusable lumen (Figure 1e). Cells were stained for nucleus and F-actin and images were subsequently used for automatic segmentation and quantification of the angiogenic sprouting (Figure 1f). The sprouting area, number of nuclei within the sprouts and sprouting distance were quantified using built-in image analysis protocols of the software of the high-content microscope. Since max projections were used for quantification, branches and nodes are not meaningful and thus not included in the quantified parameters.

Next, we optimized the concentration for sunitinib inhibition (Figure 2a, Appendix A), and show that sunitinib inhibits sprouting from concentrations >10 nM, while angiogenesis is completely inhibited at concentrations >50 nM. To characterize the signal window (SW), Z-factor (Z′) and the coefficient of variation (CV) for each parameter and select the most optimal quantification parameter (sprouting distance, nuclei in vessels and total vessel area), 0 nM of sunitinib was selected as the maximum signal (*n* = 15) and 50 nM of sunitinib as the minimum signal (*n* = 14) (Figure 2b,c). This shows that while all parameters have an acceptable Z-factor ≥0.4 and signal window >2, only sprouting distance showed an acceptable coefficient of variation of ≤20% (Table 1).

Finally, we compared the inhibition of sprouting of iPSC-ECs with primary HUVECs using sunitinib (Figure 3a,b) and 3PO (Figure 3c,d). This showed that 50% of sprouting length reduction (IC50) is achieved at sunitinib concentrations of around 20 nM (95% confidence interval (CI): 12.9–28.0) for iPSC-ECs and 66 nM (95% CI 43.70–88.58) for HUVECs. Interestingly, HUVECs still showed single cell migration into the collagen-I at high concentrations of sunitinib, whereas the sprouting of iPSC-ECs was completely blocked. Additionally, 3PO showed a significant but partial inhibition of sprouting at a concentration of 10 µM, which was similar in iPSC-ECs and HUVECs.

## 3. Discussion

We described a phenotypic angiogenesis inhibition assay which consists of 40 individually addressable, perfused micro vessels in a standardized microfluidic cell culture platform that includes physiologically relevant cues such as a three-dimensional hydrogel, flow and angiogenic gradients. The assay is shown to be robust and reproducible, showing the potential to be integrated within the drug-screening infrastructure and enables the study of the anti-angiogenic effect of compounds.

The parameters quantified in this study (sprout area, nuclei in vessels and sprout length) all showed acceptable Z’ for complex phenotype assays [23]. The quantification of sprout length had the lowest CV and was the most robust parameter. Although total sprouting area is an interesting phenotypic readout to study, as this describes the lumen development and functioning tip and stalk cells [24], we observed differences in sprouting density. Probably, these differences in sprout density are caused by differences in initial seeding densities: proper sprout development requires DLL4-Notch signaling that is only expressed in confluent monolayers [25,26].

We performed our analysis on 2D max-projection images, which reduces the spatial information to the benefit of throughput and analysis time. This approach allows the quantification of sprouting length and sprouting density. However, when number of sprouts and/or directionality of sprouts is of interest, one might give preference over 3D analysis.

We validated the usage of iPSC-ECs by directly comparing with primary ECs and tested two angiogenic inhibitors: sunitinib and 3PO. Sunitinib inhibited the sprouting of both HUVECs and iPSC-ECs at nanomolar levels, which shows that the angiogenic sprouting of iPSC-ECs is mediated through VEGFR2 signaling. Interestingly, sunitinib completely inhibited the iPSC-EC sprout formation at concentrations ≥50 nM, while HUVECs still showed single cells invading into the collagen-1 matrix at this concentration. This suggests that either the iPSC-ECs are more sensitive to VEGF or less sensitive to other angiogenic factors present (e.g., PMA, S1P, basic fibroblast growth factor).

Interestingly, our data show that 3PO inhibits the angiogenic sprouting of iPSC-ECs, which suggests that, like HUVECs, they undergo the same metabolic switch to use glycolysis as the main energy source during angiogenic sprouting [21]. While it has been shown that some endothelial cells differentiated from iPSCs have lower activity of glycolysis [27], the question remains how the culture conditions dictate the cells phenotype, and whether this effect could be attributed to the specific cues added by our culture platform. For example, it has been shown that both tip cells and non-tip cells use glycolysis as well as mitochondrial respiration for energy production, and that this balance depends on microenvironmental circumstances [28].

## 4. Materials and Methods

### 4.1. Device Preparation and Cell Culture in Microfluidic Channels

The protocol to culture endothelial cells as microvessels is described previously [13,20], with a few adaptions. HUVEC (in-house isolated with approval from the Medical Ethical Committee of the Leiden University Medical Center, Leiden, The Netherlands to use for research purposes) or iPSC-EC (NCardia, Leiden, The Netherlands) were thawed from liquid nitrogen and upon thawing resuspended in human endothelial serum free medium (HE-SFM, 11111044, Thermo Scientific, USA), centrifugated at 100 g for 5 min and resuspended to yield a concentration of 1∙107 cells/mL. For every microfluidic unit, 1 µL of cell suspension was seeded in a channel pre-coated with 10 µg/mL fibronectin (F4759-1MG, Sigma-Aldrich, The Netherlands) in Dulbecco’s phosphate-buffered saline (dPBS, 14190-094, Thermo Scientific, USA), adjacent to a patterned collagen-1 gel (3447-020-01, R&D systems, UK). The ECs were cultured for 2 days in medium supplemented with 30 ng/mL vascular endothelial growth factor-165 (450-32-10, Peprotech, USA) and 20 ng/mL bFGF (100-18b Peprotech, USA) (further referred to as vessel culture medium) to form confluent microvessels.

### 4.2. Inhibition of Angiogenic Sprouting

An angiogenic sprouting mixture was prepared by supplementing HE-SFM with 50 ng/mL VEGF, 2 ng/mL phorbol 12-myristate 13-acetate (PMA, 10-2165, Focus Biomolecules, USA) and 500 nM sphingosine-1-phosphate (S1P, S-2000-1 mg, Echelon Biosciences, USA). A total of 10 mM sunitinib stock solution in DMSO was first diluted to 25 µM in basal medium, which was then serially diluted with 0.001% DMSO used as control. A stock solution in DMSO of 150 mM of 3PO was serially diluted, with 0.007% DMSO used as control. The angiogenic sprouting mixture with inhibitors was added to the bottom perfusion inlet well and outlet well to induce angiogenic sprouting, and a vessel culture medium containing the inhibitor was added to the gel inlet and outlet and the top perfusion inlet and outlet well.

### 4.3. Fixation, Staining and Imaging

The cell culture medium was aspirated and 25 µL of 4% paraformaldehyde (PFA) in phosphate-buffered saline (J61899, Alfa Aesar, USA) was added to all the perfusion inlet and outlet wells. The device was placed under a slight angle to induce flow (e.g., by placing one side of the plate on a lid) and incubated for 10 min at room temperature. After fixation, the PFA was aspirated from the wells and the microfluidic chips were washed twice with 50 µL Hank’s balanced salt solution (HBSS, 14025-050, Thermo Scienfitic, USA) in all the perfusion inlets and outlets, followed by a permeabilization step of 0.2% Triton-X100 (VWR 28817295) and a second wash step with HBSS. Nuclei were stained using 1:2000 Hoechst 33258 (H3569, Life Technologies, USA) and F-actin using 1:200 Phalloidin-TRITC (P1951 Sigma-Aldrich, The Netherlands) in HBSS with 25 µL for every perfusion inlet and outlet well. After adding the staining, the plate was placed under a slight angle and incubated at room temperature for 30 min in the dark, followed by two wash steps with HBSS.

Images were acquired using a confocal microscope (ImageXpress Confocal Micro, Molecular Devices, USA) using a 60 µm pinhole spinning disk and 10X Plan Apo objective. Images were acquired in the DAPI and TRITC channels with shading and background subtraction. Imaging depth was set at 16 bits, binning at 1, and imaging resolution at 2048 × 2048 (0.677 µm/pixel). Autofocus was set at a 120 µm offset from channel bottom. 80 Z-steps were acquired with 1 µm Z-step interval and a total of 2 sites with 10% overlap acquired per well. Max projections were stitched in FIJI/ImageJ v1.53n using the pairwise stitching plugin [29] with the ‘linear blending’ fusion method.

### 4.4. Sprouting Quantification

The sprouting was quantified using a custom module developed in Molecular Devices MetaXpress software (MetaMorph v6.5.2.351), which segments the max projection of the angiogenic sprouts into vessels and nuclei within the vessels to extract the total vessel area, the total vessel length and the y-position of the nuclei. We quantified the average migration distance per site by extracting the average Y-position of the 10 furthest nuclei in µm minus 400 µm (based on the average absolute y-position in the image where the monolayer grows against the gel for the negative controls). For the quantification of the area, only areas containing nuclei were used.

### 4.5. Assay Performance Quantification and Plate Acceptance Criteria

The assays signal window (*SW*) and Z-factor (Z′) are defined as follows: *AVG_max_* and *SD_max_* are the mean and standard deviation of the top (maximum) signal, respectively. Similarly, *AVG_min_* and *SD_min_* are the mean and standard deviation of the bottom (minimum) signal, respectively, and *n* is the number of measurements. Then:$$SW=\frac{\left(AVG_{max}-3\frac{SD_{max}}{\sqrt{n}}\right)-\left(AVG_{min}+3\frac{SD_{min}}{\sqrt{n}}\right)\;}{SD_{max}/\surd n}$$
$$Z’=\frac{\left(AVG_{max}-3\frac{SD_{max}}{\sqrt{n}}\right)-\left(AVG_{min}+3\frac{SD_{min}}{\sqrt{n}}\right)\;}{AVG_{max}-AVG_{min}}$$
where *SW* = signal window, *Z*′ = Z-factor.

We used a recommended acceptance criterion for Z-factor ≥ 0.4, signal window ≥2, coefficient of variation (CV) ≤ 20% as acceptance criterion for the Max-signal, and *SD_min_* ≤ *SD_max_* as acceptance criteria for the Min-signal.

## 5. Conclusions

We have demonstrated that iPSC-ECs are an effective alternative to primary endothelial cells when used in a physiologically relevant *in vitro* angiogenesis inhibition screening assay. The combination of a standardized microfluidic 3D cell culture platform with a scalable and more standardized cell source is a major step in the standardization of physiologically relevant *in vitro* angiogenesis assays, as it offers the required robustness, compatibility and scalability to be integrated within the drug-screening infrastructure.

## Figures and Tables

**Figure 1 ijms-21-04804-f001:**
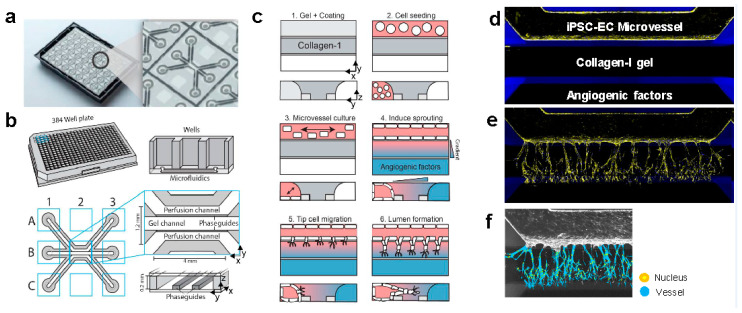
Angiogenesis assay of perfused induced pluripotent stem cell–endothelial cell (iPSC-EC microvessels. (**a**) Bottom of the microfluidic cell culture device. On the right 1 of 40 microfluidic units that are integrated underneath the 384-well plate is depicted. (**b**) Schematic overview of a single microfluidic unit/chip. The microfluidic channels are separated by ridges (‘phaseguides’), which enable the patterning of hydrogels in the central channel (‘gel channel’) while there is still contact with the adjacent channels (‘perfusion channels’). (**c**) Method to culture a microvessel within a microfluidic device and induce gradient driven angiogenic sprouting. (**d**) Microvessel 2 days after seeding iPSC-ECs as single cells. Cells form a monolayer against the patterned collagen-1 gel stained for F-actin (yellow) and nucleus (blue). (**e**) Gradient driven sprouting angiogenesis of iPSC-ECs after 2 days of stimulation with angiogenic growth factors. (**f**) Automated segmentation of vessel-like structures within the collagen-1 gel. Reproduced from van Duinen, V.; et al. Standardized and Scalable Assay to Study Perfused 3D Angiogenic Sprouting of iPSC-derived Endothelial Cells *In vitro*. *J. Vis. Exp.* 2019 [13].

**Figure 2 ijms-21-04804-f002:**
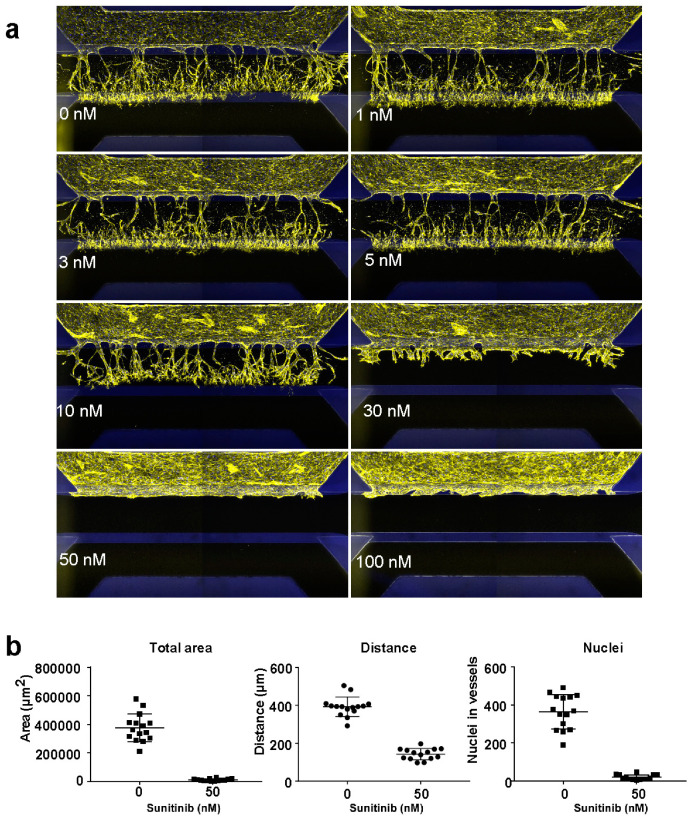
Concentration optimization for inhibition of angiogenic sprouting of iPSC-ECs using sunitinib. (**a**) Representative images of a concentration range of sunitinib. (**b**) Quantification of the vessel area, nuclei density and sprouting distance of maximal inhibition (50 nM) and no inhibition (control, 0 nM).

**Figure 3 ijms-21-04804-f003:**
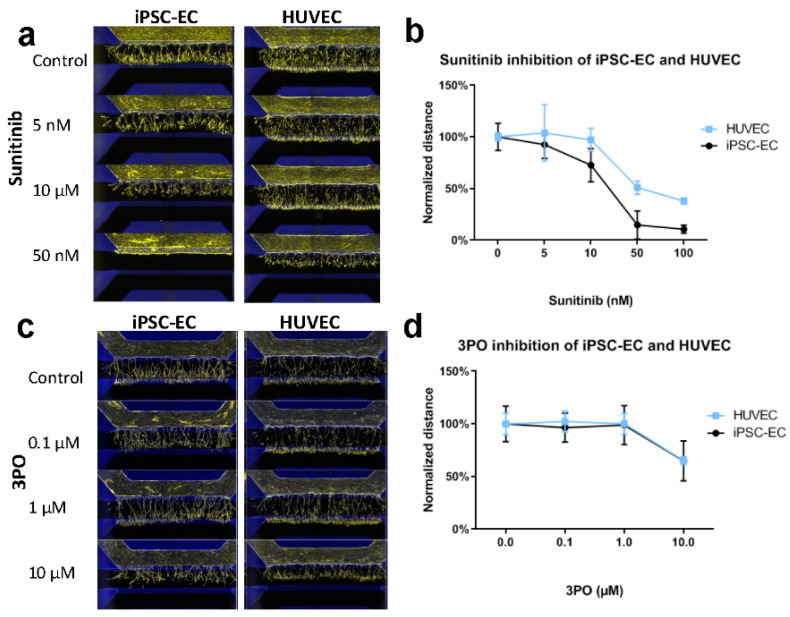
Quantification of inhibition in angiogenic sprouting of iPSC-EC and HUVEC microvessels. (**a**,**b**) Sunitinib inhibited angiogenic sprouting of both iPSC-EC and HUVEC microvessels. While sprouting of iPSC-ECs was completely inhibited at 50 nM, HUVECs still showed limited migration and sprout formation (*N* = 2). (**c**,**d**) Inhibition with 3PO shows a significant reduction in sprouting at 10 µM of both IPSC-EC and HUVEC microvessels (*N* = 2).

**Table 1 ijms-21-04804-t001:** Assay performance characteristics of the quantified parameters for iPSC-ECs. The CV at maximum (CV_max_) and minimum signal (CV_min_) are derived from 0 nM and 50 nM respectively. Recommended values as found in Iversen et al. [23].

	Total Area	Distance	Nuclei	Reference Values [23]
Signal window	11.70	14.76	11.68	>1 acceptable
Z-factor	0.78	0.75	0.77	>0.5 excellent
Assay variability ratio	0.84	0.94	0.88	<0.6 recommended
CV_max_ (%)	25	13	24	<20 acceptable
CV_min_ (%)	70	20	71	<20 acceptable

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
