# Peer review of "Robust and Scalable Angiogenesis Assay of Perfused 3D Human iPSC-Derived Endothelium for Anti-Angiogenic Drug Screening"

_ijms, 2020, doi:10.3390/ijms21134804_

Round 1

Reviewer 1 Report

The article by van Duinen et al. is well written and scientifically sound. However, this is a simple technical improvement to something already presented in the literature by the same authors or by others and the technical addition is quite predictable. Perhaps, authors failed to put in evidence the real innovation of the system.

Author Response

We would like to thank the reviewer for the critical evaluation of our manuscript and understand that the technical innovation could be seen as limited.

While we have previously shown that this platform is suitable to study angiogenesis, it is the first time that this assay is used to study the effect of anti-angiogenic compounds. Furthermore, we assess the robustness and scalability of this assay using performance metrics and demonstrate that this assay is amenable for routine drug screening. This is, in our opinion, the real innovation, as platforms/assays with comparable complexity are never used in such a screening setting as they lack the required scalability.

Therefore, we changed the title of the manuscript to "Robust and scalable angiogenesis assay of perfused 3D human iPSC-derived endothelium for anti-angiogenic drug screening", as this gives a better representation of the innovation of our work.

Reviewer 2 Report

The manuscript by van Duinen et al. is an interesting work regarding iPSC-derived ECs and microfluidics. The manuscript is well written and highlights the rationale and the main points clearly. However, the work would benefit from addressing the following points:

1) The title seems a bit descriptive. Perhaps the word "development" could be used in it.

2) The first sentence of the abstract is somewhat cumbersome to read and could be maybe divided into two sentences.

3) Figure 2 graph and table are not easily readable because of poor quality.

4) Is there any other measurements that could be done to measure angiogenesis in the system developed?

Author Response

We would like to thank the reviewer for the careful consideration of our manuscript.

1) We changed the title to

“Robust and scalable angiogenesis assay of perfused 3D human iPSC-derived endothelium for anti-angiogenic drug screening”

Not only is this less descriptive, this title better highlights the real innovation of this system in our opinion (iPSC-EC, 3D and perfusion on one hand, and robustness and scalability that enable screening on the other hand).

2) We agree and changed this sentence to:

To advance pre-clinical vascular drug research, in vitro assays that capture the process of angiogenesis are needed. Such assays should combine physiological relevant culture conditions with robustness and scalability to enable drug screening

to

To advance pre-clinical vascular drug research, in vitro assays of angiogenesis are needed that combine physiological relevant culture conditions with robustness and scalability to enable drug screening.

In our view, this now better highlights were angiogenesis assays need to be innovated: the combination of possibility to do screening while offering physiological complexity.

3) We have updated the manuscript with a high-resolution image including an increased font-size to improve readability

4) While other measurements can be done, such as 3D analysis of the angiogenic sprouts, we have chosen to analyzing max projection in this case. While losing some spatial information, this greatly reduces analysis complexity and analysis time. We have added this in our discussion:

[155] We performed our image analysis on 2D max-projection images, which reduces the spatial information to the benefit of throughput and analysis time. This approach works well for sprouting length and sprouting density. But, when number of sprouts and or directionality of sprouts is of interest, one might give preference over 3D analysis.

Round 2

Reviewer 1 Report

This new presentation of the manuscript better focuses the goal of the authors. However, to fully sustain their development of a reliable method for anti-angiogenic screening, at least another drugs in addition of Sunitinib should be used.

Author Response

We would like to thank the reviewer for his/her helpful suggestion of adding another inhibitor. However, as shown in Figure 3, we tested already another compounds in addition to Sunitinib: 3PO, both on HUVEC as well as on iPSC-ECs. Also, as both compounds act through different mechanisms, this sufficiently supports our results in our opinion.

Round 3

Reviewer 1 Report

In my opinion, the use of  3-(3-pyridinyl)-1-(4-pyridinyl)-2-propen-1-one (3PO) is not appropriate, as it is a general inhibitor of glucose metabolism, acting in different cell types. It has an anti-angiogenic effect but could not be considered another anti-angiogenic compound.

I would propose instead bevacizumab, as it is used already in the clinics.